# Outcomes of Prospectively Followed Pregnancies in Rheumatoid Arthritis: A Multicenter Study from Romania

**DOI:** 10.3390/life13020359

**Published:** 2023-01-28

**Authors:** Anca Bobircă, Anca Angela Simionescu, Anca Emanuela Mușetescu, Cristina Alexandru, Florin Bobircă, Mihai Bojincă, Andra Bălănescu, Mihaela Micu, Codrina Ancuța, Romina Sima, Laura Andreoli, Ioan Ancuța

**Affiliations:** 1Department of Internal Medicine and Rheumatology, “Carol Davila” University of Medicine and Pharmacy, 050474 Bucharest, Romania; 2Department of Internal Medicine and Rheumatology, Dr. I. Cantacuzino Clinical Hospital, 011437 Bucharest, Romania; 3Department of Obstetrics and Gynecology, Filantropia Hospital, “Carol Davila” University of Medicine and Pharmacy, 050474 Bucharest, Romania; 4Rheumatology Department, University of Medicine and Pharmacy of Craiova, 200638 Craiova, Romania; 5Department of General Surgery, “Carol Davila” University of Medicine and Pharmacy, Dr. I. Cantacuzino Clinical Hospital, 011437 Bucharest, Romania; 6Department of Internal Medicine and Rheumatology, “Sfanta Maria” Hospital, 011172 Bucharest, Romania; 7Rheumatology Division, 2nd Rehabilitation Department, Rehabilitation Clinical Hospital, 400066 Cluj-Napoca, Romania; 8Rheumatology Department, Grigore T. Popa University of Medicine and Pharmacy, 700115 Iasi, Romania; 9Department of Obstetrics and Gynaecology, The “Bucur” Maternity, “Saint John” Hospital, “Carol Davila” University of Medicine and Pharmacy, 050474 Bucharest, Romania; 10Rheumatology and Clinical Immunology Unit, Spedali Civili, Department of Clinical and Experimental Sciences, University of Brescia, 25121 Brescia, Italy

**Keywords:** rheumatoid arthritis, adverse pregnancy outcomes, small for gestational age, pre-eclampsia, corticosteroids, pregnancy

## Abstract

Women with rheumatoid arthritis (RA) may carry an increased risk of adverse pregnancy outcomes (APO). The aims of this study were to compare pregnancy outcomes in RA patients as compared to the general obstetric population (GOP) and to identify a risk profile in RA. A case-control study was conducted on 82 prospectively followed pregnancies in RA and 299 pregnancies from the GOP. The mean age at conception was 31.50 ± 4.5 years, with a mean disease duration of 8.96 ± 6.3 years. The frequency of APO in RA patients was 41.5%, 18.3% experienced spontaneous abortions, 11.0% underwent preterm deliveries, 7.3% had small for gestational age infants, 4.9% experienced intrauterine growth restriction, 1.2% experienced stillbirth, and 1.2% suffered from eclampsia. The risk of APO was correlated with a maternal age higher than 35 years (*p* = 0.028, OR = 5.59). The rate of planned pregnancies was 76.8%, and the subfertility rate was 4.9%. Disease activity improved every trimester, and approximately 20% experienced an improvement in the second trimester. Planned pregnancies and corticosteroids use (≤10 mg daily) were protective factors for APO in RA pregnancies (*p* < 0.001, OR = 0.12, *p* = 0.016, OR = 0.19, respectively). There was no significant association between APO and disease activity or DMARDs used before and during pregnancy. Regarding the comparison between the RA group and the controls, RA mothers were significantly older (*p* = 0.001), had shorter pregnancies (*p* < 0.001), and had neonates with a lower birth weight (*p* < 0.001).

## 1. Introduction

Rheumatoid arthritis (RA) is a chronic, autoimmune, inflammatory disease affecting the joints and musculoskeletal structures. Indeed, RA can induce progressive disability if not treated properly. The prevalence of this disease in European countries varies from 0.31% to 0.9% of the population, with a man:woman ratio of 1:3 [1]. The incidence of RA increases with age, from 8.7 per 1000 between 18 to 34 years to 36.2 per 1000 between 35 to 44 years [2]. The condition affects lifestyle, consecutive to rheumatic symptoms, immunosuppressive treatment, and psychological issues, which can limit a patient’s desire for family planning [2]. The number of women with autoimmune diseases including RA becoming pregnant is increasing [3], as is the maternal age for conception [4]. The time to conceive for women with RA is longer if patients are older or nulliparous, have higher disease activity, and if they use nonsteroidal inflammatory drugs or daily prednisone administration >7.5 mg [5].

Since the beginning of the 1990s, the link between estrogen levels and the severity of induced arthritis has been demonstrated [6,7]. Pregnancy had a beneficial effect on RA, because the disease activity diminishes when the levels of female sex hormones are high [8], while there is an increased risk of postpartum flare [9]. A recent meta-analysis of 237 pregnancies in RA patients, using objective scores of disease activity, has demonstrated that only 60% of patients improve in disease activity during pregnancy, and 50% experience worsening symptoms in the postpartum period [10]. Active disease was reported in approximately 35 to 50% of pregnant women with RA [11].

Autoimmune diseases represent a risk factor for adverse pregnancy outcomes, as well as autoimmune disease activity. The more severe type of RA (characterised by the presence of rheumatoid factor RF and anti-citrullinated protein antibodies, ACPAs) is indicative of higher active RA, more extensive joint destruction, and extraarticular manifestation. Study population reports have shown a correlation between RA, disease activity, and pregnancy complications, such as premature rupture of membranes, low birth weight, and intrauterine growth restriction, stillbirths, eclampsia, and increased risk of caesarean section [12,13,14].

Pregnancy outcomes can be severely influenced by disease activity and immunosuppressive treatment during the preconception period and gestation. An inactive disease before conception usually remains stable during pregnancy [15]. Another important factor that needs to be considered when planning a pregnancy in RA patients is the therapy, as not all the disease-modifying anti-rheumatic drugs (DMARDs) are compatible with pregnancy. According to the latest recommendations from the American College of Rheumatology [16], methotrexate, leflunomide, mycophenolate mofetil, and cyclophosphamide must be stopped 1–3 months before conception, being the most teratogenic therapeutic formulas. As for bDMARDs, rituximab, and anti-TNFα agents, especially certolizumab, are safe for pregnancy usage during the first and second trimester. The remaining biological medications lack enough information to support their usage during pregnancy [16]. Nonsteroidal anti-inflammatory drugs (NSAIDs) are usually recommended for symptom control, being safe for pregnant women or women who are trying to conceive. There is evidence that long-term use of NSAIDS, particularly during the periovulatory period, can affect fertility; therefore, the rheumatologist must take this into account when prescribing NSAIDS [17].

In this paper, we aimed to describe the adverse pregnancy outcomes (APO) among pregnancies in RA patients and disease course during conception for these women, identifying characteristics that were associated with APO. As a secondary objective, we compared the frequency of APO in the RA cohort to that of a control group.

## 2. Materials and Methods

### 2.1. Design and Setting

An observational, multicentered case-control study of RA patients who became pregnant was conducted, specifically at the Rheumatology Department of the Dr. I. Cantacuzino Hospital in Bucharest, but also in four other clinics across the country, namely at the Department of Internal Medicine and Rheumatology, Dr I. Cantacuzino Clinical Hospital and Department of Internal Medicine and Rheumatology, “Sfanta Maria” Hospital from Bucharest, Rheumatology Department, University of Medicine and Pharmacy of Craiova, Rheumatology Division, 2nd Rehabilitation Department, Rehabilitation Clinical Hospital Cluj-Napoca, and the Rheumatology Department, Grigore T. Popa University of Medicine and Pharmacy, Iasi. Inclusion criteria were women older than 18 years old at conception, with an RA diagnosis before pregnancy, who became pregnant between 2012–2022. The exclusion criterion was voluntary termination of pregnancy for either medical or personal reasons. Although the inclusion criteria were fulfilled for 98 cases, in the final study, 82 pregnancies were included in the case group, since 16 pregnancies (16.3%) were excluded due to the mother’s choice to have an abortion. The study included 82 pregnancies from 59 women with RA, as follows: 1 patient had 4 pregnancies, 4 patients had 3 pregnancies, and 12 patients had 2 each, with the rest having only 1 pregnancy. Due to the chronic disease’s heterogeneity, which presented itself differently during conception, we treated each pregnancy as a distinct case. The local ethics committee gave its approval (Study Code 5981/8 April 2022), and written consent for all patients was obtained. Patients were examined by a rheumatologist and obstetrician 3 months before conception, throughout pregnancy, and 3 months post-partum. At every visit, data regarding disease activity measured by DAS28CRP (a laboratory and clinical international score), rheumatoid arthritis therapy, and pregnancy evolution and outcomes were collected. Based on pregnancy outcomes, we divided the subjects into two subgroups, comparing disease aspects between them, defined as adverse pregnancy outcome (APO) and no adverse pregnancy outcome (No APO). The APO group included preterm delivery (before 37 weeks of gestation), small for gestational-age infant (SGA), intrauterine growth restriction (IUGR), eclampsia, spontaneous abortion, and stillbirth. A fetus was defined small for gestational age when its estimated fetal weight or abdominal circumference (on ultrasound evaluation) was below the 10th percentile for the gestational age reference ranges. Abnormal fetal Doppler evaluation associated with estimated fetal weight or abdominal circumference below the 10th percentile defined fetuses with intrauterine growth restriction [18]. The definition of pre-eclampsia (PE) is based on the new onset of hypertension (blood pressure ≥140 mmHg systolic or ≥90 mmHg diastolic) and significant proteinuria (≥300 mg/24 h) or with evidence of renal, hepatic, or hematological dysfunction associated hypertension after 20 weeks of gestation [19,20]. Eclampsia is the convulsive manifestation of the hypertensive disorders of pregnancy [21]. Spontaneous abortion is the loss of pregnancy naturally before viability established at twenty weeks of gestation, with stillbirth being defined as occurring after twenty weeks of gestation [22]. Birth weight under or equal of 2500 g is defined as low birth weight (LBW), according to the World Health Organization [23].

### 2.2. Case and Control Definitions

Case-control patients were 299 consecutive women over 14 years old admitted to Filantropia Clinical Hospital in 2014 for delivery. The data from the control group were collected in 2014, within research grant 20062, from Carol Davila University of Medicine ad Pharmacy.

### 2.3. Variables

Demographic variables were collected for cases and control patients enrolled. For RA patients, we collected variables 3 months before conception, throughout pregnancy, and 3 months post-partum, such as disease activity measured by DAS28CRP (a laboratory and clinical international score), rheumatoid arthritis therapy, pregnancy evolution and outcomes, as well as the immunological risk factors and the prevalence of other autoimmune diseases as secondary Sjögren’s syndrome, Hashimoto’s thyroiditis, and antiphospholipid syndrome. In order to define a planned pregnancy, the patients underwent a pregnancy planning consultation. The rheumatologist advised the best RA medication compatible with pregnancy, limiting the NSAIDs intake in periovulatory period, testing for antiphospholipid antibodies and anti-RO antibodies, as well as postponing the procreation process for women with an active disease.

Disease activity was measured using DAS28(CRP) 3 months before pregnancy and at every trimester during pregnancy. A DAS28(CRP) value below or equal to 3.2 (remission and low disease activity) was considered as inactive disease, whereas a value greater than 3.2 (moderate and high disease activity) was defined as an active RA. A RA flare was defined as progressing to a more advanced disease stage based on the DAS28(CRP), considering a more active disease. The number of flares during pregnancy and 3 months postpartum were registered.

The information collected on the management of RA included the type of medication, either biological DMARDs or synthetic DMARDs, 3 months prior to conception, and in each trimester.

### 2.4. Statistical Analysis

The system used for descriptive and analytical statistics was IBM SPSS Statistics version 20.0. Values of categorical variables were registered as numbers and percentages while continuous variables were presented as mean and standard deviation (SD), median, and minimum and maximum, respectively, depending on the normal distribution. To determine any significant associations, the data were analyzed using Student’s *t*-test and the Mann–Whitney U test with a two-tailed hypothesis for continuous data, and the Pearson Chi-square and Fisher’s exact test for categorical data, respectively. Having more than one pregnancy per patient, for odds rations (OR) and 95% confidence interval (CI), we used generalized estimating equation analyses adjusted for age. For all tests, a significant statistical result was considered at *p*-value < 0.05.

## 3. Results

### 3.1. Cohort Characteristics

The RA cohort’s characteristics are illustrated in Table 1. The mean age at conception was 31.50 ± 4.5 years, with a mean disease duration of 8.96 ± 6.3 years, while the percentage of women over 35 years was 18.3%. In the studied group, 76.8% (63 patients) had a planned pregnancy, with proper adjustment of RA treatment. The time to pregnancy was between 0 to 172 weeks (TTP), with a median of 8 weeks. On the other hand, analyzing the planned pregnancies, excluding those with TTP 0, the median TTP was 12 weeks, while 6.3% (4 patients) had a TTP greater than 12 months. There were 66 live births from 66 patients, with the mean pregnancy length being 37.8 ± 1.7 weeks.

The RA seropositivity rate (rheumatoid factor and/or ACPA positivity) was very high in our group, with a value of 86.6%. Out of the 37 patients who were tested for the auto-antibodies anti-Ro/SSA and anti-La/SSB, 3 were positive, and they were closely monitored for congenital heart block throughout the pregnancy, with only one pregnancy presenting a spontaneous abortion.

The antiphospholipid antibodies were found in 10% of cases (4 patients from the 40 examined), and all of them followed individualized treatment with low-dose aspirin in monotherapy or associated with heparin therapy according to the latest EULAR recommendations [16], taking into consideration the history of obstetrical or thrombosis events. The outcomes of those patients were as follows: one patient had a spontaneous abortion, one patient had preterm birth, and two patients had uneventfully pregnancy.

Thirty-six patients (54.5%) gave birth by cesarean section. The mean neonate’s birth weight was 2961.8 ± 508.4 g; low birth weight was found in 15.5% of cases.

During pregnancy, only 14.6% of RA patients had a flare, but 72.7% suffered disease relapse in the first 3 months post-partum.

Analyzing the RA activity using DAS28(CRP), an inactive disease was more common before pregnancy, as well as during each trimester. Moreover, when compared to the preconception period, disease activity improved every trimester by a maximum of roughly 20% in the second trimester.

The case group included 82 pregnancies from 59 patients diagnosed with RA, of which 80.5% (N = 66) had a live birth, 18.3% (N = 15) had a spontaneous abortion, and 1.2% (N = 1) had a case of stillbirth (Table 2). Based on the pregnancy outcomes, 58.5% (N = 48) had a good pregnancy outcome, without any maternal or fetal complications, and 41.5% (n = 34) had an adverse pregnancy outcome including preterm delivery, SGA, IUGR, spontaneous abortion, stillbirth, or eclampsia (Table 2). The frequency of premature deliveries was 11% when related to the entire cohort, and 13.6% of all live births registered in the study.

One of the premature infants was diagnosed with a congenital cardiac malformation, namely a bicuspid aortic valve, with normal heart function, and also with a congenital nasal dermoid cyst. In this particular case, the RA mother experienced rheumatoid vasculitis and was treated with rituximab for 5 years, with the last infusion being one year before conception, with no specific anti-rheumatic drugs taken during the preconception period or during the pregnancy.

There was only one case of eclamsia in the third trimester, in a 31-year-old primiparous, non-obese woman with spontaneous pregnancy. The patient had a RA therapeutic regime compatible with pregnancy, treated with low-dose aspirin prophylaxis for a hereditary thrombophilia with a low risk of thrombosis, levothyroxine for Hashimoto’s disease and sulfasalazine 2 g daily for RA control, with an inactive disease under this medication during pregnancy. She was diagnosed with gestational diabetes at 28 weeks of gestation and high blood pressure, proteinuria, edema, and seizures recognized as eclampsia at 38 weeks of gestation. The patient gave birth by cesarean section to a 3090 g healthy infant, with an APGAR score of 8/10.

In our cohort, only one woman needed assisted reproductive technology, at 35 years old with a good pregnancy outcome. She was diagnosed with infertility 5 years before RA diagnosis; she had an inactive rheumatic disease before conceptions and during pregnancy, being under treatment with 10 mg of oral prednisone, until 12 weeks of gestations, without immunosuppressive therapy.

During the 3 months before conception, 32.9% were without any RA treatment, the biologic DMARDs being used preferentially, in 17.1% of women (Table 3). The bDMARDs prescribed were rituximab and anti-TNFα inhibitors, such as etanercept, infliximab, and adalimumab. All three cases of patients treated with rituximab had the last infusion in the first trimester of pregnancy, before performing a pregnancy test. Methotrexate was prescribed in 12.2% of cases during the preconception period; five patients stopped the MTX therapy 3 months prior to gestation. Leflunomide treatment was registered in 6.1% of cases, only one of whom received the washout procedure with cholestyramine, without any delivery or fetal complications. The other four cases had a miscarriage.

In contrast, during pregnancy (Table 4), the treatment most used to control the disease activity was glucocorticoids (GCs), with a frequency of 22.0%. The GC dosages were as follows: 11 patients received low dosages, orally (equal to or under 7.5 mg of oral prednisone or equivalent per day), 5 patients received 10 mg of oral prednisone or equivalent per day and 1 patient received a single dose of 250 mg methylprednisolone intravenously, at 20 weeks of gestation for an RA flare. Additionally, for 3 patients, dexamethasone (6 mg × 4 doses/12 h) was used at 32 weeks of gestation to accelerate fetal lung maturation, without any bad impact on the pregnancy outcome.

Regarding the biological therapy during pregnancy, 17.1% (N = 14) were treated with anti-TNF inhibitors and rituximab, with eight of them discontinuing the therapy at a positive pregnancy test result, at around 4 weeks of gestation. Five subjects required biological treatment until the second trimester (four with etanercept and one with infliximab). There was one case of certolizumab initiation in the second trimester due to insufficient control of the disease activity. This patient had a high disease activity in the preconception period and in the first trimester, being under oral prednisone (10 mg daily). Ultimately, she had a preterm delivery at 34 weeks of gestation, with an IUGR fetus, the birth weight of the neonate being 1800 g.

The only case of stillbirth at 21 weeks of gestation was an unplanned pregnancy in an obese patient, who received MTX and infliximab in the first trimester (with an inactive disease before and during gestation), without any other associated conditions.

### 3.2. Comparing the Two Subgroups

The two subgroups were mostly similar in terms of demographic features. Even though there was no significant difference in the average age at conception, women older than 35 years were more likely to have an adverse pregnancy outcome (29.4% vs. 10.4%, *p* = 0.028, OR (95%CI) = 5.59 (1.53–23.08)). An important finding was the fact that planned pregnancies were statistically significantly associated with no adverse pregnancy outcome (91.7% vs. 55.9%, *p* < 0.001, OR (95%CI) = 0.12 (0.03–0.43)). Hence, the duration from decision-making to conception was shorter for the APO group, with more than half of the subjects having a median time to pregnancy of 0, with a borderline signification (*p* = 0.052), but this result could be biased due to the high percentage of unplanned pregnancies in the APO subgroup.

When comparing the pregnancy length between the APO and No APO subgroups, there is a significant difference with a lower value for APO mothers, *p* < 0.001.

For the 66 live births, infants from the APO subgroup had a significantly lower birth weight (2388.9 ± 527.0 vs. 3176.7 ± 290.2, *p* < 0.001, OR (95%CI)= 1.01 (1.00–1.02)). All 10 neonates with LBW were from mothers with pregnancy complications, as follows: three preterm deliveries, three small for gestational age infants, and four with IUGR. Evaluating the birth weight while comparing the planned and unplanned pregnancies, a significantly higher percentage of neonates with LBW had RA mothers with unplanned pregnancies (40.0% vs. 10.7%, *p* = 0.037).

Furthermore, we analyzed the impact of the disease activity before delivery on birth weight, comparing the infants ≤2500 g and those with a weight >2500 g by mother’s disease activity in the third trimester. The results showed that only 3 cases out of 10 with LBW infants had an active condition in the third trimester (*p* = 1.000). On the other hand, there was no association between GC therapy and low birth weight (*p* = 0.715).

Similarly, there was no link between disease activity before and during pregnancy with adverse pregnancy outcomes. However, out of the 15 cases of spontaneous abortion, 10 (66.7%) of them had an active disease before conception, and 8 (53.3%) had an active disease during the first trimester.

The treatment administered 3 months before and during pregnancy was compared between the two cohorts and the results showed no difference regarding the use of disease-modifying anti-rheumatic drugs, neither synthetic nor biological. A significant result was regarding the intake of orally corticosteroids during pregnancy, namely prednisone ≤10 mg daily. This increased the chance of having a live birth with no maternal or fetal complications (*p* = 0.016, OR (95%CI) = 0.19 (0.04–0.94)). There were five patients treated with MTX during the first trimester of pregnancy, and three of them had a spontaneous abortion and one had a stillbirth. The fifth had no adverse outcomes.

### 3.3. Comparing with Controls

Regarding the control group, there were five twin pregnancies; thus, we included 304 neonates in the control group. When comparing the 66 births registered in RA patients with a control group (Table 5) the results showed a significantly higher age at conception for RA mothers, 31.3 ± 4.4 y.o vs. 29.2 ± 5.5 y.o (*p* = 0.001). The frequency of APO is higher for RA patients (27.3% vs. 19.1%), without a significant *p*-value. Here, C-section delivery was preferred in the RA group compared to the controls, without statistical significance. The neonates from RA mothers had a lower birth weight compared to controls (2961.8 ± 508.4 vs. 3236.2 ± 562.4, *p* < 0.001), with LBW being more frequent in the RA cohort, with borderline *p*-values, *p* = 0.064. Furthermore, pregnancy length is significantly shorter for RA patients, with premature deliveries being more common (13.6% vs. 8.7%); however, there is no statistical significance. In terms of adverse pregnancy outcomes, SGA was slightly more prevalent in the RA group, while IUGR was more frequent among controls but not statistically significant (*p* = 0.303, *p* = 0.852, respectively). There were no cases of eclampsia or pre-eclampsia found in the control group.

## 4. Discussion

Herein, we described a sample of 82 pregnancies in patients diagnosed before conception with RA, analyzing the potential factors that were associated with adverse pregnancy outcomes. Our results showed a delay in family planning when it comes to those women suffering from a chronic disease, such as RA, concordant to other authors [4]. Moreover, older females (>35 years old) in our study had a higher probability of having a pregnancy with complications, consistent with data from general pregnant populations [24]. Time to pregnancy is used to establish subfertility, defined as a TTP greater than 12 months. In our study, 4.9% of patients (N = 4) of the entire cohort (6.3% from the subgroup of planned pregnancies) had subfertility problems according to the TTP. However, Brouwer et al., in a study on 254 patients, showed that subfertility was present in a higher percentage of RA mothers, in 42% of cases [5].

According to our findings and in line with the literature, a carefully planned pregnancy, taking into account disease activity and, in particular, potential risks for malformations induced by RA medication, is critical for enhancing the likelihood of avoiding adverse pregnancy outcomes [25,26].

In our study, 16.3% of women required a voluntary termination of pregnancy, while in a Romanian statistical analysis from 2019, the rate of fertile women who required this procedure was 10.9‰ [27].

Our results showed that miscarriages, preterm deliveries, small for gestational age infants, and intrauterine growth restriction were the most common complications, findings which are in accordance with other studies [28,29]. Moreover, when compared to the control group, prematurity and SGA were more frequent in RA patients in our study, while in a larger study IUGR and premature rupture of membrane had a higher prevalence than in the general population [30].

A more specific entity was that one mother in our population developed eclampsia at 38 weeks of gestation, a very rare condition defined as high blood pressure, proteinuria, edema, and seizures. To our knowledge, this is one of the few cases of eclampsia in a rheumatoid arthritis pregnancy described in the literature.

The rate of cesarean section for RA group was 54.4%, the frequency being higher as compared to the control group. This result is in accordance with other studies [28,30].

Miscarriage had the largest percentage when analyzing the adverse pregnancy outcomes. Moreover, out of the 15 cases of spontaneous abortion, more than half did not benefit from medical preconception counselling. Consequently, in some of the cases, the RA medication before conception was not in accordance with the recommendation at that time. A further significant result in this paper was the association between unplanned pregnancy and lower weight at birth.

In our study, only 14.6% of patients had a flare during pregnancy, and it did not have a great impact on the pregnancy outcome, results that are very different from the findings of a previous study of 46 RA pregnancies between 2001 and 2009, in which more than half of the subjects had at least one flare [31]. Concerning the rate of post-partum flares, our results showed a high prevalence among RA patients enrolled, but there was no association with adverse pregnancy outcomes.

The risk factors for disease relapse after birth is a controversial topic, and there is not sufficient data to establish exactly where the disease activity during pregnancy or if the type of treatment has a definite effect on post-partum flares [10,32].

Even though pregnancy can improve RA, one of the most significant factors to consider when planning a pregnancy is disease activity, which can be linked to both a long time to conception and a poor pregnancy prognosis [33]. In our study, over half of the subjects had an inactive disease before and during pregnancy, with an increased proportion during the second and third trimesters. For improving pregnancy outcomes, it is desired to achieve RA remission before conception, since it was discovered that low disease activity at the beginning of the pregnancy is more likely to remain stable until birth [15]. In some studies, the disease activity, mainly in early pregnancy correlated with adverse pregnancy outcomes, especially lower birth weight and preterm delivery [34,35]. Nevertheless, in this paper, there was no significant association between disease activity before and during pregnancy and preterm birth, IURG, small gestational age infants, spontaneous abortion, or lower birth weight.

The treatment to be chosen for childbearing women with RA must not only be effective in controlling the condition but also safe and without any adverse effects on fertility and pregnancy. In the past few years, it was demonstrated that some of the anti-rheumatic drugs do not have a negative impact on pregnancy and if properly managed, the maternal and fetal risks are reduced to a minimum [36]. In our study, there was no significant association between the therapy administered before and during pregnancy with adverse outcomes. The recent recommendation has contraindicated the use of methotrexate and leflunomide therapies during the 3 month preconception period and throughout gestation, not only in women but also in men with RA, even though there is not sufficient data to prove the teratogen effect in a rheumatology practice [26,37]. Our results showed that four out of five patients treated with MTX or LEF had adverse outcomes, particularly spontaneous abortion and one case of stillbirth under methotrexate. Another essential aspect of these findings is that most of the patients treated with those therapies had an unexpected pregnancy, so once again we can emphasize the importance of a planned conception and prenatal counseling for these women.

In the past decade, biologic disease-modifying anti-rheumatic medication has notably improved the RA patient’s quality of life. Anti-TNFα agents are the most studied among the biological therapies not only for women but for men with autoimmune diseases, and they can be considered safe when used during the preconception period, as well as in the early stages of pregnancy [38]. However, some studies have concluded that this biological treatment can increase the risk of postpartum infections and, according to some authors, they should be discontinued at 30 weeks of gestation [39,40]. In our research, the patients enrolled in this study underwent therapies with anti-TNFα agents, such as etanercept, adalimumab, certolizumab and infliximab during gestation, but also rituximab in the first trimester without any difference when comparing the two subgroups. All cases of anti-TNF inhibitors were stopped in the second trimester, except for one that received treatment in the second and third trimester until 34 weeks of gestation and developed polyhydramnios and preterm premature rupture of membranes.

Glucocorticoids (GCs) were more frequently used during pregnancy than in the 3 months prior to conception, with most of the patients treated with GCs in the third trimester (11 out of 18 patients taking ≤10 mg of oral prednisone or equivalent, daily). There was a significant difference between the two subgroups regarding steroid intake during pregnancy, showing that this therapy improved the chance of having a pregnancy without adverse outcomes. Recent studies have demonstrated that oral GCs have no teratogenic effect when used for pregnant patients [41], but a correlation was made with IUGR and prematurity, with the risk increasing with higher GC dosages [34,42,43]. Nonetheless, in our study, the patients were treated with lower or equal to 10 mg of oral prednisone daily, and there was no significant correlation between GC usage and neonates with low birth weight.

This study is the first prospective study on RA pregnancies in Romania, and as there has not been much research on this subject at the national level up until now, we hope that our analysis will provide crucial information about how pregnancy affects RA and vice versa for women in our country, providing doctors with more confidence in managing the disease. Given that the study is multicentric, the primary limitation is the lack of homogeneity in the interdisciplinary follow-up of pregnancies. Another limitation is the small number of patients who enrolled. There are patients included with two or more pregnancies, which can be a bias, as the fear and stress regarding RA activity in relation to pregnancy evolution can be higher for the first pregnancy. Further investigation is required to provide a more thorough description of the illness and its consequences on pregnancy and postpartum disease evolution.

## 5. Conclusions

In this study, adverse pregnancy outcomes are more frequent for RA pregnancy than the control group. Moreover, RA women have a higher age at conception, a shorter pregnancy length, and neonates with lower birth weight. The disease activity improved during pregnancy in 20% of cases and no teratogenicity was noticed. There was no correlation between disease activity or anti-rheumatic modifying disease drugs administration and pregnancy outcomes. The APO mothers had a shorter pregnancy length and newborns with lower birth weights compared with the No APO mothers. Unplanned pregnancy and age older than 35 years at conception were the major risk factors for APO in the studied population. Corticotherapy was a protective factor for uneventful pregnancy. Additional research with a larger cohort size is needed to investigate risk factors for adverse pregnancy outcomes in RA patients, as well as to establish the disease course post-partum and long-term outcome in children born to RA mothers.

## Figures and Tables

**Table 1 life-13-00359-t001:** Cohort characteristics.

Characteristics	All Patients N = 82	No APO N = 48 (58.5%)	APO N = 34 (41.5%)	*p*-Value
Age at RA diagnosis (years) mean ± SD	22.74 ± 7.4	23.04 ± 7.0	22.32 ± 7.9	0.847
Age at conception (years) mean ± SD	31.50 ± 4.5	31.15 ± 4.3	32.0 ± 4.9	0.398
Age at conception > 35 years old	15 (18.3%)	5 (10.4%)	10 (29.4%)	0.028
Disease duration (years) mean ± SD	8.96 ± 6.3	8.25 ± 6.2	9.97 ± 6.5	0.228
Juvenile idiopathic arthritis n (%)	18 (22.0%)	11 (22.9%)	7 (20.6%)	0.802
Rheumatoid arthritis n (%)	64 (78.0%)	37 (77.1%)	27 (79.4%)	0.802
Planned pregnancy n (%)	63 (76.8%)	44 (91.7%)	19 (55.9%)	<0.001
TTP (weeks) median (min–max) mean ± SD	8 (0–172) 16.45 ± 26.2	10 (0–172) 19.27 ± 29.2	0 (0–96) 12.47 ± 20.9	0.052
Smoking n (%)	18 (22.0%)	8 (16.7%)	10 (29.4%)	0.170
Positive RF and/or ACPA n (%)	71 (86.6%)	41 (85.4%)	30 (88.2%)	0.712
Anti-Ro/SSA and anti-La/SSB antibodies n (%); N = 37	3 (8.1%)	2 from 22	1 from 15	
Antiphospholipid antibodies n (%); N = 40	4 (10%)	2 from 23	2 from 17	
Hashimoto’s disease n (%)	3 (3.7%)	1 (2.1%)	2 (5.9%)	0.567
Sjögren’s syndrome n (%)	3 (3.7%)	2 (4.2%)	1 (2.9%)	1.000
Sedentary lifestyle n (%)	31 (37.8%)	18 (37.5%)	13 (38.2%)	0.946
Higher education n (%)	44 (53.7%)	27 (56.2%)	17 (50.0%)	0.576
Obesity n (n%)	5 (6.1%)	1 (2.1%)	4 (11.8%)	0.155
BMI before pregnancy mean ± SD	21.67 ± 3.5	21.94 ± 2.9	21.3 ± 4.2	0.101
BMI at delivery mean ± SD	25.02 ± 4.3	25.3 ± 3.9	24.5 ± 4.8	0.228
Nulliparity before RA diagnosis n (%)	71 (86.6%)	40 (83.3%)	31 (91.2%)	0.348
Flare during pregnancy n (%)	12 (14.6%)	9 (18.8%)	3 (8.8%)	0.342
C-section n (%) N = 66	36 (54.5%)	23 (47.9%)	7 (38.9%)	0.512
Pregnancy length (weeks) mean ± SD N = 66	37.8 ± 1.7	38.4 ± 0.9	36.2 ± 2.2	<0.001
Low birth weight (≤2500 g) n (%); N = 66	10 (15.5%)	0	10 (55.6%)	<0.001
Neonates’ birth weight (g) mean ± SD N = 66	2961.8 ± 508.4	3176.7 ± 290.2	2388.9 ± 527.0	<0.001
RA flare post-partum n (%); N = 66	48 (72.7%)	36 (75.0%)	12 (66.7%)	0.498
Active RA before pregnancy	39 (47.6%)	23 (47.9%)	16 (47.1%)	0.939
Active RA in the first trimester	28 (34.1%)	15 (31.2%)	13 (38.2%)	0.511
Active RA in the second trimester N = 67	19 (28.4%)	13 (27.1%)	6 (31.6%)	0.713
Active RA in the third trimester N = 66	20 (30.3%)	13 (27.1%)	7 (38.9%)	0.353

Abbreviations are as follows: n—number; %—percentage; RA—rheumatoid arthritis; SD—standard deviation; TTP—time to pregnancy; min—minimum; max—maximum; RF—rheumatoid factor; ACPA—anti-citrullinated protein/peptide antibody; BMI—body mass index; *p* < 0.05 is statistically significant.

**Table 2 life-13-00359-t002:** Pregnancy outcomes.

Definition		Frequency, N	Percent from Entire Cohort N = 82	Percent from Live Births N = 66
No adverse pregnancy outcome (No APO)		48	58.5%	72.7%
Adverse pregnancy outcome (APO)	Preterm infants	9	11.0%	13.6%
	Severe preterm < 32 W	1		
	Moderate preterm (32–34 W)	3		
	Late preterm (35–36 W)	5		
	Small for gestational age	6	7.3%	9.1%
	Intrauterine growth restriction	4	4.9%	6.1%
	Miscarriage (<20 weeks of gestation)	15	18.3%	
	Stillbirth	1	1.2%	-
	Eclampsia	1	1.2%	1.5%

**Table 3 life-13-00359-t003:** Medication for 3 months before conception.

Treatment	All Patients N = 82	No APO N = 48 (58.5%)	APO N = 34 (41.5%)	*p*-Value
MTX n (%)	10 (12.2%)	5 (10.4%)	5 (14.7%)	0.559
SSZ n (%)	12 (14.6%)	7 (14.6%)	5 (14.7%)	0.988
HCQ n (%)	3 (3.7%)	2 (4.2%)	1 (2.9%)	1.000
LEF n (%)	5 (6.1%)	1 (2.1%)	4 (11.8%)	0.155
NSAIDs n (%)	10 (12.2%)	6 (12.5%)	4 (11.8%)	0.920
GCs n (%)	9 (11.0%)	7 (14.6%)	2 (5.9%)	0.294
bDMARDs n (%)	14 (17.1%)	8 (16.7%)	6 (17.6%)	0.907
TNF inhibitors	11 (13.4%)	6 (12.5%)	5 (14.7%)	
Rituximab	3 (3.7%)	1 (2.1%)	2 (0.1%)	
Drug-free n (%)	27 (32.9%)	16 (33.3%)	11 (32.4%)	0.926

Abbreviations are as follows: n—number; %—percentage; MTX—methotrexate; SSZ—sulfasalazine; HCQ—hydroxychloroquine; LEF—leflunomide; NSAIDs—nonsteroidal anti-inflammatory drugs, GCs—glucocorticoids; bDMARDs—biologic disease-modifying antirheumatic drug; *p* < 0.05 is statistically significant.

**Table 4 life-13-00359-t004:** Medication during pregnancy.

Treatment	All Patients N = 82	No APO N = 48 (58.5%)	APO N = 34 (41.5%)	*p*-Value
MTX n (%)	5 (6.1%)	1 (2.1%)	4 (11.8%)	0.155
HCQ n (%)	4 (4.9%)	2 (4.2%)	2 (5.9%)	1.000
LEF n (%)	5 (6.1%)	1 (2.1%)	4 (11.8%)	0.155
SSZ n (%)	11 (13.4%)	6 (12.5%)	5 (14.7%)	0.773
NSAIDs n (%)	6 (7.3%)	2 (4.2%)	4 (11.8%)	0.226
GCs n (%)	18 (22.0%)	15 (31.2%)	3 (8.8%)	0.016
bDMARDs n (%)	14 (17.1%)	8 (16.7%)	6 (17.6%)	0.907
TNF inhibitors	11 (13.4%)	6 (12.5%)	5 (14.7%)	
Rituximab	3 (3.7%)	1 (2.1%)	2 (0.1%)	
Drug-free n (%)	33 (40.2%)	21 (43.8%)	12 (35.3%)	0.442

Abbreviations are as follows: n—number; %—percentage; MTX—methotrexate; SSZ—sulfasalazine; HCQ—hydroxychloroquine; LEF—leflunomide; NSAIDs—nonsteroidal anti-inflammatory drugs, GCs—glucocorticoids; bDMARDs—biologic disease-modifying antirheumatic drug; *p* < 0.05 is statistically significant.

**Table 5 life-13-00359-t005:** Comparison between RA births and controls.

	RA Group N = 66	Control N = 299	*p*-Value
Age at conception (years) mean ± SD	31.3 ± 4.4	29.2 ± 5.5	0.001
APO n (%)	18 (27.3%)	58 (19.1%)	0.135
Preterm n (%)	9 (13.6%)	26 (8.7%)	0.527
<32 W	1	5	
32–34 W	3	6	
35–36 W	5	15	
IUGR n (%)	4 (6.1%)	20 (6.6%)	0.852
Small for gestational age n (%)	6 (9.1%)	17 (5.6%)	0.303
Eclampsia n (%)	1 (1.2%)	0	
C-section n (%)	36 (54.5%)	136 (45.5%)	0.182
Pregnancy length (weeks) mean ± SD	37.8 ± 1.7	38.5 ± 1.9	<0.001
Neonates’ birth weight mean ± SD	2961.8 ± 508.4	3236.2 ± 562.4	<0.001
Low birth weight ≤2500 g n (%)	10 (15.2%)	24/304 (7.9%)	0.064

Abbreviations are as follows: n—number; %—percentage; RA—rheumatoid arthritis; SD—standard deviation; *p* < 0.05 is statistically significant.

## Data Availability

Not applicable.

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
