# Peer review of "Outcomes of Prospectively Followed Pregnancies in Rheumatoid Arthritis: A Multicenter Study from Romania"

_life, 2023, doi:10.3390/life13020359_

Round 1

Reviewer 1 Report

Bobirca and colleagues analysed pregnancies in women with rheumatoid arthritis and investigated adverse pregnancy outcomes, medication and disease course. Furthermore, outcomes of a control group were presented. The topic is of interest and various results are presented in the manuscript. However, the focus of the work is not clear and due to methodological issues, I would recommend a major revision.

My main recommendation is to focus on one main aspect to be reported. Reading the different parts of the manuscript, it is not clear, what the main objective was to be investigated. It feels more like everything which was captured in the study, now also needs to be reported in the manuscript. This makes it difficult to the reader to get the main message. The manuscript is very comprehensive and in parts redundant (e.g., variables are described in 2.1 and 2.3), and it would profit from a stringent presentation.

My second main issues concerns the methods. From the description, I was not able to understand the study design and several questions arose:

-       - Were pre-defined case report forms used to capture date?

-        - Ethical approval was gained in 2022, but data was already collected in 2012. This needs an explanation. Is there an additional ethical approval from earlier on?

-        - Does the ethical approval also refers to the control group?

-        - The study is multicentric. How many centres participated?

-        - The study is prospective. Can I presume, that all patients were enrolled before pregnancy? Also the controls?

-        - Control group is not described appropriately, what are the inclusion and exclusion criteria? Did they have the same follow-up times?

-        - All of the women in the control group were enrolled in 2014, is that correct? I presume a high bias due to the big time gap in cases in controls. This should be added to tehe discussion.

-        - A total of n=82 women were enrolled in the 10 years of the study. Were there any other women enrolled, but excluded for this analysis?

-        - Did every women only contribute one pregnancy?

Investigating pregnancies always comes with the problem of choosing the correct denominator which can be, e.g., total number of women, pregnancies, foetuses, neonates, but also live birth depending on the outcome to be investigated. To my opinion, the denominator was not chosen correctly at every stage of the analysis. E.g., percentages of preterm birth should not be related to the abortion, since these pregnancies did not have the “chance” to have a preterm birth. I would recommend to decide for every outcome, which denominator is correct and also to state it in the manuscript.

Further questions regarding methodology:

-        - Were all of the outcomes and variables available for all pregnancies?

-        - Were there any loss-to-follow-ups during the study period?

-        - Odds ratios are provided in the results, but the statistical method is not mentioned in the methods. Please add more information. Are the odds ratios unadjusted? If yes, why was no adjusted regression analysis performed. This should be added to the limitations. If no, please indicate covariables.

My recommendation would also be to use the STROBE checklist for the reporting of the study results.

Further minor comments:

-       -  Abbreviations: abbreviations are introduced but not applied throughout the manuscript or re-introduced again; Nonsteroidal anti-inflammatory drugs can be abbreviated earlier in the introduction; HQC must be changed to HCQ (e.g., table 3)

-        - Line 93: reference 9 does not seem to be appropriate

-        - Line 136/137: Information on twin pregnancy etc. belongs to the results

-      -  SGA calculation: Please indicate the reference curves which were used

-       - Flares: how were flares identified?

-        - Characteristics of the study cohort is usually presented in table 1 – before presenting the outcomes.

-        - How was “planned pregnancy” defined?

-        - bDMARDs: TNF inhibitors and Rituximab should be named separately in the tables 3 and 4. When were the last infusions of rituximab prior/ during pregnancy?

-        - Results section is in part very detailed on a patient level. Maybe detailed information can be added to a supplement.

Author Response

Dear Reviewer,

We are very thankful to you for the pertinent notes; we have carefully read the comments and have revised/ completed the manuscript accordingly. Our responses are given in a point-by-point manner, in the attached word. As well, all the changes to the manuscript are highlighted in red, using track changes options.

Kind Regards,

Cristina Alexandru, 

Corresponding author (maria-cristina.steopoaie@rez.umfcd.ro)

Reviewer 2 Report

  Thank you for requesting  to provide a review of this article, which has a subject of high interest. 

   The main purpose of the analysis was to compare pregnancy outcomes in Rheumatoid Arthritis (RA) as compared to the general obstetric population and to identify a risk profile in this kind of disease.

   The main question adressed by the research is whether pregnancy-related complications are specifically associated with Rheumatoid Arthritis and also if adverse pregnancy outcomes are more frequent for RA pregnancies, than in the control group. 

   The study is a prospective observational multicentered cohort study, on 82 prospectively followed pregnancies in RA and 299 pregnancies from the general obstetric population. The topic is original and relevant in the field and brings usefull knowledge regarding the subject. A comprehensive search strategy was used  and included obstetrical population in general and also pregnant women with RA and the review methodology was comprehensive with screening and data extraction. When it comes to the methodology used, no specific improvements should be considered from my point of view.

   The conclusions are consistent with the evidence and the arguments presented, and they adress properly to the main question which conducted the analysis and brings sufficient data for the ideas that adverse pregnancy outcomes are more frequent for RA pregnancies, women have a higher age at the conception due to the therapy used for the disease, neonates from women with RA have a lower weight at birth etc. Of great importance is the fact that corticotherapy was a protective factor for uneventfull pregnancies. 

   The references are appropriate and well suited for this kind of study. 

    Regarding the tables and figures used in the article, they provide suitable information about the cases and show significant statistical references. They are also well understandable and the information is easy to be followed. There are no other comments required about these items, from my point of view.

  Regarding the structure and accuracy of the phrases, the manuscript has well structured information, with supported evidence and well structured phrases.

   The manuscript is original and well defined. The results provide an advance in current knowledge. The results are being interpreted appropriately and are significant, as well as the conclusions.

  The article is written in an appropriate way. 

  The study is correctly designed and the analysis is being performed at high standards, so the data are robust enough to draw the conclusion. 

   Surely the paper will attract a wide readership. 

   The English language is appropriate and well understandable.

   There are a few things to add in the lines below, but the article should be published after the corrections are made: 

Line 53: only 1 space between „man:women”

Line 62: it was demonstrated, not „is has been demonstrated”

Line 66: „,” after „activity”

Line 67: that only 60%, not „only 60%”

Line 67: are worsening, not „worsening”

Line 94: we aimed to describe, not „we aimed at describing”

Line 95: and to identify, not „and identifying”

Line 102: multicentered, not „multicenter”

Line 110: close the „()” after „2022”

Line 167: fetal, not „foetal”

Line 201: who were tested, not „who tested”

Author Response

Dear Reviewer,

We are very thankful for your kind review; we have carefully read the comments and have corrected the manuscript according to your suggestions, we attached a word document with our response as well.

We hope that in this new form, the manuscript will be suitable for publication.  

Kind regards!

Cristina Alexandru,

Corresponding author: maria-cristina.steopoaie@rez.umfcd.ro

Round 2

Reviewer 1 Report

The authors have addressed all of my comments from the first revision. I still recommend a major revision since the statistical methods applied were not appropriate. This is due to the fact, that major information was missing in the first version of the manuscript, e.g. number of women does not equal the number of pregnancies, no information on major statistics in the methods.

Justification: The authors applied multiple logistic regression adjusted for age to calculate the risk for adverse pregnancy outcomes. Since one women could contribute several pregnancies to the analysis, it can be expected that the outcomes of one woman are correlated (e.g., the risk for preeclampsia is increased if a women was already diagnosed with preeclampsia in a previous pregnancy) and therefore, appropriate statistical methods should be applied, e.g. clustered analysis, generalised estimating equation analyses [1, 2].

Minor comments:

-        Table 2: Frequency of Small for gestational in the live births should be checked.

-        Table 3/ Table 4: “anti-TNF inhibitors” is not correct. It is either TNF-inhibitor or anti-TNF treatment

References

1.            Tsai YC, Chang HC, Chiou MJ, Luo SF, Kuo CF. Fetal-neonatal and maternal pregnancy outcomes in women with rheumatoid arthritis: a population-based cohort study. BMJ open. 2022 Oct 26; 12(10):e059203.

2.            Redeker I, Strangfeld A, Callhoff J, Marschall U, Zink A, Baraliakos X. Maternal and infant outcomes in pregnancies of women with axial spondyloarthritis compared with matched controls: results from nationwide health insurance data. RMD Open. 2022 Jul; 8(2).

Author Response

Dear Reviewer,

Thank you for taking the time to review our work, we appreciate your recommendations. We changed the statistical test, as you suggested. The word document detailing the adjustments is attached.

Your feedback is greatly appreciated, and we hope that this form is as close as possible to your recommendations.

Cristina Alexandru
